# Facial mimicry interference reduces working memory accuracy for facial emotion expressions

Emil Holmer[1,2]*, Jerker Rönnberg[1,2], Erkin Asutay[1,3], Carlos Tirado[1], Mattias Ekberg[1]

**1** Department of Behavioural Sciences and Learning, Linköping University, Linköping, Sweden, **2** Linnaeus Centre HEAD, Linköping University, Linköping, Sweden, **3** JEDI Lab, Linköping University, Linköping, Sweden

* emil.holmer@liu.se

**Data Availability Statement:** The data file (.csv) and analysis script (in R) for this study are available on the Open Science Framework (see: https://osf.io/yhq47/?view_only=9d22e7feb6e043829c1735bc191fbc44).

## Abstract

Facial mimicry, the tendency to imitate facial expressions of other individuals, has been shown to play a critical role in the processing of emotion expressions. At the same time, there is evidence suggesting that its role might change when the cognitive demands of the situation increase. In such situations, understanding another person is dependent on working memory. However, whether facial mimicry influences working memory representations for facial emotion expressions is not fully understood. In the present study, we experimentally interfered with facial mimicry by using established behavioral procedures, and investigated how this interference influenced working memory recall for facial emotion expressions. Healthy, young adults ($N = 36$) performed an emotion expression $n$-back paradigm with two levels of working memory load, low (1-back) and high (2-back), and three levels of mimicry interference: high, low, and no interference. Results showed that, after controlling for block order and individual differences in the perceived valence and arousal of the stimuli, the high level of mimicry interference impaired accuracy when working memory load was low (1-back) but, unexpectedly, not when load was high (2-back). Working memory load had a detrimental effect on performance in all three mimicry conditions. We conclude that facial mimicry might support working memory for emotion expressions when task load is low, but that the supporting effect possibly is reduced when the task becomes more cognitively challenging.

## Introduction

Working memory is the cognitive system used when we are engaged in an activity and try to deal with both internal and external input streams, from which we need to select the relevant bits and suppress the irrelevant, when we solve the task [1]. In everyday communication, this system is critical not only for making meaning out of verbal input, but also to integrate verbal utterances with information in non-verbal expressions [2], such as emotion expressions [3]. In optimal conditions, working memory underlies the effective use of multimodal and multilevel

**Funding:** This work was partly supported by Linnaeus Centre HEAD excellence center grant (349-2007-8654) from the Swedish Research Council (https://www.vr.se) and by a program grant (2012-1693) from FORTE (https://forte.se/), both awarded to JR. The funders had no role in study design, data collection and analysis, decision to publish, or preparation of the manuscript.

**Competing interests:** The authors have declared that no competing interests exist.

abstractions [4,5]. The aim of the present study was to investigate whether working memory for facial emotion expressions is affected by behavioral interference of presumed facial mimicry, when working memory load is low and high.

According to resource models of working memory, the amount of information to be stored and the precision at which this information is internally represented determines successful processing [4–6]. In line with this perspective, increased demand on working memory storage consistently leads to poorer memory performance [1,7], and performance is poorer when stimuli are abstract [8,9] or of low quality [10,11]. Negative effects on working memory performance have also been reported for facial emotion expressions [12–16]. Further, when working memory resources are occupied, the otherwise prioritized processing of emotions might be suppressed [17,18], and the recognition of emotional content might be impaired [19–21]. Thus, working memory resources seem to be critical for the processing of emotional input.

From another perspective, numerous studies show that the recognition of emotional states when working memory demands are low is influenced by humans' tendency to imitate other's facial expressions (i.e., facial mimicry) [22–32]. Thus, when the cognitive system allows for it, facial mimicry might be a prioritized route for emotion recognition. It might be the case, that facial mimicry contributes to the precision of the working memory representation of facial emotion expressions. However, it remains unclear whether facial mimicry supports the processing of emotional face expressions regardless of the level of working memory load.

Simulation, i.e., the reenactment of an observed facial emotion expression in the observer [33], may contribute to the observer's recognition of an emotion expression [34–36]. Thus, it influences the precision of representations of emotion expressions. It has been known for some time that watching faces expressing different emotion expressions elicits facial mimicry responses [37]. This finding makes the notion that facial feedback contributes to the accurate representation of emotion expressions plausible. The strongest evidence for this contribution comes from experimental work that has shown that facial mimicry interference impairs [25,26,28,29] or slows down [27,31] perception of emotion expressions from faces. Others have reported enhanced perception [32] and short-term memory [24] of happy emotion expressions, when provoking smiling in participants. In addition, concurrent movement of facial muscles, suppressing facial mimicry, seems to impair emotion categorization precision [30]. Further evidence for the role of facial mimicry in the recognition of emotion expressions comes from a study of alexithymia, which is the impaired ability to represent, recognize, and verbally label emotional states. The study suggests that this population has weaker facial mimicry responses than healthy controls when watching emotion expressions [38]. On a related note, a recent meta-analysis showed that facial feedback (i.e., using facial movement to provoke a certain emotional response in yourself) has a small but detectable effect on emotional experience [39]. However, studies of medical conditions that lead to facial paralysis (i.e., Moebius syndrome), and thus the inability to produce facial mimicry responses, have shown preserved emotion recognition ability [40], suggesting that facial mimicry is not critical to successfully represent emotions. Alternative perspectives on the role of facial mimicry regard it as a communicative signal that is selectively used as a means of communication, depending on the context of the interaction [41–43]. Further, studies suggest that cognitive load might suppress facial mimicry responses and that facial mimicry is not critical for the processing of emotion expressions in demanding settings [44,45], such as when working memory load is high.

To the best of our knowledge, the existing studies investigating the effect of facial mimicry manipulations in relation to working memory processing have done so in short-term memory tasks with a low level of working memory load. For example, Kuehne et al. [24] observed that short-term memory for happy emotion expressions improved when smiling was provoked by asking participants to engage in a pen-between-teeth procedure (i.e., participants had to bite

on a pen, without touching the pen with their lips). In another study, Sessa, Lomoriello, and Luria [46] investigated the effect of facial mimicry interference on short-term memory storage of facial emotion expressions, and the neural signature of the effect was measured using electroencephalography. Although facial mimicry interference did not influence accuracy in their change-detection short-term memory task, a decreased amplitude in an event-related potential component usually seen as a marker of visual working memory was observed in participants with medium to high levels of affective empathy. These earlier findings suggest that not only perception of emotion expressions might be influenced by facial mimicry, but perhaps also short-term and working memory storage. However, previous studies did not manipulate both working memory load and facial mimicry interference, and the relationship between these mechanisms is thus not known.

According to Wood, Rychlowska, et al. [35], interfering with facial mimicry produces somatosensory feedback that is incongruent with simulation of the emotion expression. This in turn leads to poorer precision of the representation. Distorted feedback reduces the speed and accuracy of perception. Wood, Lupyan, et al. [23] proposed that incongruent facial mimicry may interfere with the precision of visual working memory for emotion expressions. Wood, Rychlowska et al. further proposed that somatosensory simulation may extend visual working memory capacity for facial expressions. These arguments suggest that simulation and facial feedback might impact the precision of working memory for facial emotion expressions and that a negative effect of facial mimicry interference on the precision of representations might become more profound when working memory load is high. There are few working memory models focusing on how emotional input is stored in working memory. One model assumes that a dedicated cognitive structure (i.e., the hedonic detector) is used for evaluating the valence and intensity of an input that is represented and stored in working memory [47–49]. This structure has a neutral valence point from which positive or negative deviations are identified, and this information is used to form a representation of the specific emotion. Thus, if we assume that facial mimicry is related to the emotional experience of an observer [39], it follows that the precision of the representation will be reduced by facial mimicry interference. This assumption, together with the resource model perspective on working memory [5,6], inspired our expectation that task load and mimicry interference will produce interacting negative effects in the context of working memory for facial emotion expressions.

## The present study

We sought to examine whether interfering with facial mimicry negatively affects memory precision of facial emotion expressions when working memory load was low and high. To investigate this, we used an *n*-back working memory task [50]. The task is commonly used to investigate working memory in experimental settings [1,7,51]. The paradigm involves memory storage and updating as well as executive control, thus, it is designed to deplete the working memory system. In everyday communication, multiple sources of information are tracked simultaneously to build a coherent representation of the meaning of the interaction [5]. This complexity is well simulated in a resource-demanding task such as the *n*-back task. In a typical design of the task, the participant is required to keep track of a sequence of items (e.g., digits, words, pictures), and to make a response if the current item matches the item *n* steps back in the sequence [1,7,50]. By manipulating *n*, often from one to three, working memory resources are increasingly taxed.

For the purpose of the present study, we used one (low, 1-back) and two (high, 2-back) item working memory loads in a facial emotion expressions *n*-back task. The targets were happy, neutral, and angry expressions corresponding to the positive-negative range on an

assumed internal valence scale [47]. The *n*-back paradigm has been used to investigate working memory for facial emotion expressions in some previous studies [12–14,52,53]. Results indicate that working memory load has a detrimental effect on performance. Thus, representations of facial emotion expressions are, like other stimuli, vulnerable to working memory load. Because of this, we limited working memory load to two items. Piloting indicated that three-item working memory load might be too difficult [16]. Most typically, *n*-back tasks are episodic and each item is an exact match to a previous item. However, in some versions of the task, responses are made based on one feature of the stimulus at the same time as other features are suppressed. In the present study, we wanted to investigate the precision of memory representations of emotion expressions and not the exact episodic memory trace. Therefore, we prepared an *n*-back task where the participant had to monitor the invariance of facial expressions across different faces [54,55]. We predicted that performance accuracy would decrease with increasing working memory load. We expected this effect to be the strongest when facial mimicry interference was high.

## Methods

### Participants

Native Swedish-speaking individuals with normal or corrected vision and between 18 and 35 years old were recruited for this study. All participants met the following inclusion criteria: absence of disabilities including sensory and physical disabilities but excluding corrected visual deficits, absence of neuropsychiatric and developmental disabilities, and absence of psychological disorders. The sample size was determined based on resource limitations and heuristics [56]. More specifically, previous studies with similar manipulations and designs showed statistically significant effects with samples in the range of 20 to 50 participants [22,24,28,29,57] and such an approach was also deemed feasible in our study given the available funding and the planned balancing of conditions. Thirty-six participants were recruited, and our expected sample size was reached. Participants were recruited by advertisement at the University campus, and by approaching groups of students to inform them about the study. The age of the participants spanned between 20 and 28 ($M = 23.7$, $SD = 1.99$), and most were female (72%). The Matrices subtest from WAIS-IV [58] indicated better-than-normal non-verbal ability ($M = 13.1$, $SD = 3.08$), and the Letter-number sequencing subtest from WAIS-IV revealed typical, or slightly higher, levels of verbal working memory ($M = 11.4$, $SD = 2.25$). The study adheres to the ethical principles of the Declaration of Helsinki [59], was approved by the Regional Ethical Review Board in Linköping (dnr 2017/141-31), and the participants gave their written informed consent for participation. The individual displayed in Fig 1 in this manuscript has given written informed consent (as outlined in PLOS consent form) to publish these case details. Data was collected in the spring of 2019 and pseudonymized before analysis.

### Materials

DmDx version 5.3.1.13 [60] was used for the *n*-back task and for the physical matching practice task [61] which we implemented to familiarize participants with the response procedure in the *n*-back task. The stimuli, taken from The Radboud Faces Database [62], consisted of a total of 27 pictures depicting angry, happy, and neutral facial emotion expressions, from nine different persons. Pictures were presented with a resolution of 800 by 600 pixels in the center of the screen on a white background. We used photographs of individuals between 18 and 35 years old to match the age span to that of the potential participants and consequently minimizing the own-age bias in face processing known to impact attention-distraction [63] and memory [64]. The persons depicted in the pictures were all female, since evidence suggests that women

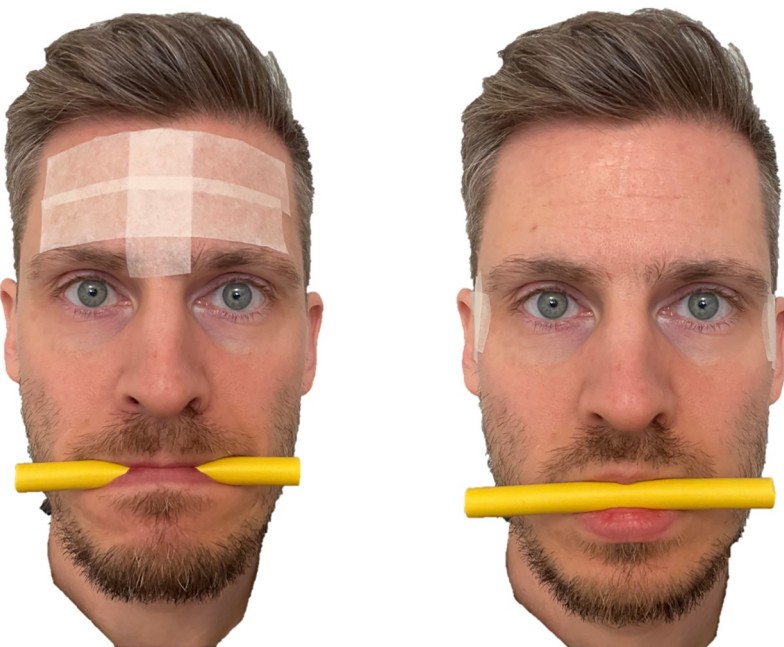

**Fig 1. Depiction of the facial mimicry interference manipulations.** Graphic example of the high interference (left) and low interference (right) facial mimicry manipulations. The individual in this manuscript has given written informed consent (as outlined in PLOS consent form) to publish these case details.

show slightly more emotional expressivity than men, especially for positive emotions and internalized negative emotions [65]. For the facial mimicry manipulations applied while performing the *n*-back task, we used plastic foam rods which were approximately 9 mm in diameter and two different sizes of a strong, non-elastic sports support tape, 4 by 2.5 cm in width. The tape is coated with a zinc oxide latex-free adhesive which is hypoallergenic.

## Design

**Facial mimicry manipulations.** In the high interference condition (see Fig 1), participants were instructed to pull back their lips over their teeth and bite down on a foam plastic rod with a constant moderate level of pressure [57]. We implemented a taping procedure modelled on previous studies [31,66], along with the use of the plastic foam rod. The purpose of the manipulations was to interfere with the activity of the zygomaticus major (by biting the foam rod) and corrugator supercilia (the taping) muscles that activate when expressing happiness through smiling and anger through frowning [37,67]. Based on previous research [31,57,66], we hypothesized that these procedures would interfere with facial mimicry involving muscles in the upper and lower part of the face, leading to facial feedback that was incongruent with sensorimotor simulation.

In the low interference condition (see Fig 1), participants were instructed to merely let the plastic foam rod rest between their lips without applying any pressure [57]. A taping procedure, based on the control condition from Wood et al. and Carpenter and Niedenthal studies [31,66], was implemented along with the use of the foam plastic rod manipulation. We reasoned that this procedure would not interfere with frowning, and that it would slightly interfere with smiling. To ensure adherence to the manipulations, participants were shown the pictures in Fig 1 and were given some time to adjust to appropriately follow the instructions.

| | Block 1 | Block 2 | | Block 3 | Block 4 | | Block 5 | Block 6 |
|---|---------|---------|---|---------|---------|---|---------|---------|
| | **No interference** | | Change of facial mimicry manipulation | **Low interference** | | Change of facial mimicry manipulation | **High interference** | |
| | 1-back | 2-back | | 1-back | 2-back | | 1-back | 2-back |

**Fig 2. Illustration of the structure of the *n*-back experiment.** An overview of how the facial mimicry interference (no, low, and high interference) and working memory load (low, 1-back, and high, 2-back) conditions were distributed across the six experimental blocks, each including 54 trials, for one participant. The order of conditions was balanced across participants.

Participants were informed that the purpose of the manipulations was to control for facial muscle tension [25].

**N-back experiment.** Participants performed six blocks of the *n*-back task. They performed either a block of 1-back (low working memory load) first and then a block of 2-back (high working memory load), or the reverse, under each of three conditions of facial mimicry interference: no interference, low interference, and high interference (see Fig 2, for an illustration of the structure of the experiment). The order of facial mimicry manipulation and working memory load was counterbalanced across the participants. Within a block, nine of the 27 pictures from the Radboud Faces Database [62] were used, representing three individuals depicting the three emotion expressions. The specific pictures used were balanced across participants and conditions. Each block started with the presentation of a cue, i.e., "1-back" or "2-back", for 4 seconds, after which a fixation cross was on screen for 1 second before the presentation of the first trial. Blocks consisted of 54 trials (i.e., pictures of facial emotion expressions), 18 trials of each emotion category (i.e., happy, neutral, and angry), and in each trial, a picture of a facial expression was displayed for 1.25 seconds, which was followed by a fixation cross presented for 0.75 seconds, before the presentation of the next trial. Response times for a trial extended until the display of the next trial, thus, the response window on each trial was 2 seconds. The participant had to respond whether the facial emotion expression matched the facial emotion expression *n* steps back (1-back or 2-back) in the sequence by pressing designated buttons for "yes" and "no". Pictures that matched on an emotion expression *n* step back never matched on the identity of the person in the picture. The three facial emotion expressions occurred as *n*-back targets an equal number of times, *n* = 6, in every sequence and the order of expressions presented was balanced. Between the two-block sequences representing a sequence for any facial mimicry manipulation, participants were given the opportunity to pause briefly (no longer than a few minutes) before continuing to the next block. Between blocks within any facial mimicry manipulation, there was a five-second pause. Before every second block, the material used for mimicry manipulation conditions was changed.

**Valence and arousal ratings.** After finishing the *n*-back task, any remaining tape was removed, and participants viewed the 27 pictures that were included in the task, one after another in a fixed random order, and rated the valence and arousal of the expressions. The concepts of valence and arousal were demonstrated through pictures of self-assessment manikins [68], and assessed on a 9-point Likert scale, from 1 (negative/weak) to 9 (positive/strong). The main purpose of the valence and arousal ratings was to validate the perceived emotional content of the pictures. The ratings also provided estimates of the participants' internal representation of the stimuli set, which we used as control variables in the analysis.

## Statistical analysis

To investigate our main prediction that a negative effect of facial mimicry on emotion expression precision increases with increasing working memory load, we performed a generalized

linear mixed effects model with the within-group factors specified as working memory load (two levels: low, 1-back, and high, 2-back) and facial mimicry interference (three levels: no, low, and high), as well as their interaction. The dependent variable was whether the response on a given trial was correct. Control factors included the block order to account for potential habituation and mnemonic effects, as well as individual mean valence and mean arousal estimates for the three types of emotional expressions in the stimuli material (angry, happy, and neutral) to model individual differences along these dimensions. The random effect structure included correlated intercepts and slopes for load at the level of the individual. Analysis was performed in R statistical software [69] using the glmer function from the lme4 package [70] for model estimation. For testing the simple effects of interactions, the emmeans package [71] was used. To deal with the binary outcome variable, a logit-link function was applied. The model was estimated using maximum likelihood estimation with the bobyqa optimizer and 1000 iterations. Satterthwaite approximation of degrees of freedom was applied to test fixed effects. Working memory load was dummy-coded, using one variable with low load as the reference at 0. Facial mimicry interference was also dummy-coded, using two variables, one for low interference and another for high interference, with no interference as the reference at 0 in both. We used the package sjPlot [72] to run comprehensive regression diagnostics for our main model, including assessments for variance, homoscedasticity, normality, random effects, and outliers, which consistently validated the model's conformity to the assumptions. To test the validity of the emotion expressions in the pictures, mean valence and mean arousal ratings for each of the categories angry, neutral, and happy expressions were compared, in two separate analyses. The random effect structure in these analyses included random intercepts at the level of the individual. A more complex random effect structure created convergence issues. The assumption of normality was violated for the rating data since it showed clustering at several areas of the curve. Hence, a Gaussian Mixture Model (GMM) approach was used for these analyses. This approach was applied by using the package mixtools [73] and the function normalmixEM to generate the best fit. It allowed the model to capture complex patterns in the data (non-linear) and perform with accuracy despite the assumption of normality being violated. The effects of angry and happy emotion expressions were modelled using two dummy-variables with neutral emotion expressions as the reference at 0. The model was estimated using maximum likelihood estimation with the Nelder-Mead optimizer and 1000 iterations. No data was missing, and the significance level was set to $\alpha = .05$ in all statistical analyses. The data file (.csv) and analysis script (in R) for this study are available on the Open Science Framework (see: https://osf.io/yhq47/?view_only=9d22e7feb6e043829c1735bc191fbc44).

## Procedure

Testing was conducted in a silent room at the University with only the test administrator and a participant present. To familiarize the participant with the response procedure of the *n*-back experiment (responding yes and no after making a decision based on the features of a stimulus), a simple physical matching task [61,74] was performed. In the physical matching task, pairs of letters that either matched or did not match were presented, and the task was to indicate with a button press when a pair matched. The physical matching task was followed by practice trials on the *n*-back emotion expressions task, which consisted of two shorter sequences with 1-back and 2-back working memory load but without any facial mimicry manipulation. Participants practised until they achieved a total error rate of 30% or less in a sequence. We assumed that potential baseline differences between participants would be negated through the implementation of the practice trials before the experiment.

**Table 1. Results from the generalized linear mixed model of correct responses.**

| Predictors | Correct response | | |
|---|---|---|---|
| | Odds Ratios | CI | p |
| (Intercept) | 31.00 | 18.04–53.28 | <0.001 |
| Load | 0.27 | 0.21–0.35 | <0.001 |
| High Interference | 0.55 | 0.35–0.86 | 0.009 |
| Low Interference | 0.94 | 0.59–1.52 | 0.812 |
| Valence | 1.08 | 1.06–1.11 | <0.001 |
| Block order | 1.08 | 1.05–1.11 | <0.001 |
| Arousal | 1.30 | 1.09–1.54 | 0.003 |
| Load * High interference | 1.35 | 1.04–1.76 | 0.024 |
| Load * Low interference | 1.02 | 0.77–1.33 | 0.912 |
| Random Effects | | | |
| $\sigma^2$ | | 3.29 | |
| $\tau_{00 \text{ participants}}$ | | 1.34 | |
| $\tau_{11 \text{ participants|Load}}$ | | 0.20 | |
| $\rho_{01 \text{ participants}}$ | | -0.85 | |
| ICC | | 0.13 | |
| N $_{\text{participants}}$ | | 36 | |
| Observations | | 11664 | |
| Marginal $R^2$ / Conditional $R^2$ | | 0.10 / 0.22 | |

CI is the 95% confidence interval while $p$ is the significance level. $\Sigma^2$ is the within participants variance and $\tau_{00}$ the between participants variance. ICC is the interclass-correlation coefficient.

## Results

### Accuracy

For accuracy, there was a main effect of working memory load ($\beta$ = -1.30, df = 11652, $p <$ .001), which means that the higher the load the lower the performance accuracy (see Table 1). Further, there was a negative main effect of high interference, $\beta$ = -0.61, df = 11652 $p$ = .009, but no main effect of low interference, suggesting that performance accuracy in the high interference condition only was poorer than in the no interference condition. We found positive main effects of valence ($\beta$ = 0.078, df = 11652, $p <$ .001) and arousal ($\beta$ = 0.26, df = 11652, $p$ = .003), which means that performance accuracy improved when the mean valence and mean arousal were higher. Finally, the main effect of block order was also positive and statistically significant ($\beta$ = 0.076, df = 11652, $p <$ .001), therefore, participants improved as they progressed in the task. However, the critical test of our study was the interaction between working memory load and high interference. Surprisingly, the statistically significant interaction was in the opposite direction of our prediction ($\beta$ = 0.30, df = 11652, $p$ = .024, displayed in Fig 3). This result suggests that the effect of mimicry interference was suppressed, rather than enhanced, by increased working memory load. As expected, the interaction between working memory load and low interference was not statistically significant (see Table 1). Following up the simple main effects on the statistically significant interaction between working memory load and high interference, revealed that when working memory load was high (2-back), the effect of high interference was not significant ($\beta$ = 0.00, df = 11652, $p$ = 1.00). The pattern and direction of the main and interaction effects suggest that there was a negative effect of the high interference condition when working memory load was low, but not when the load was high (see Fig 3).

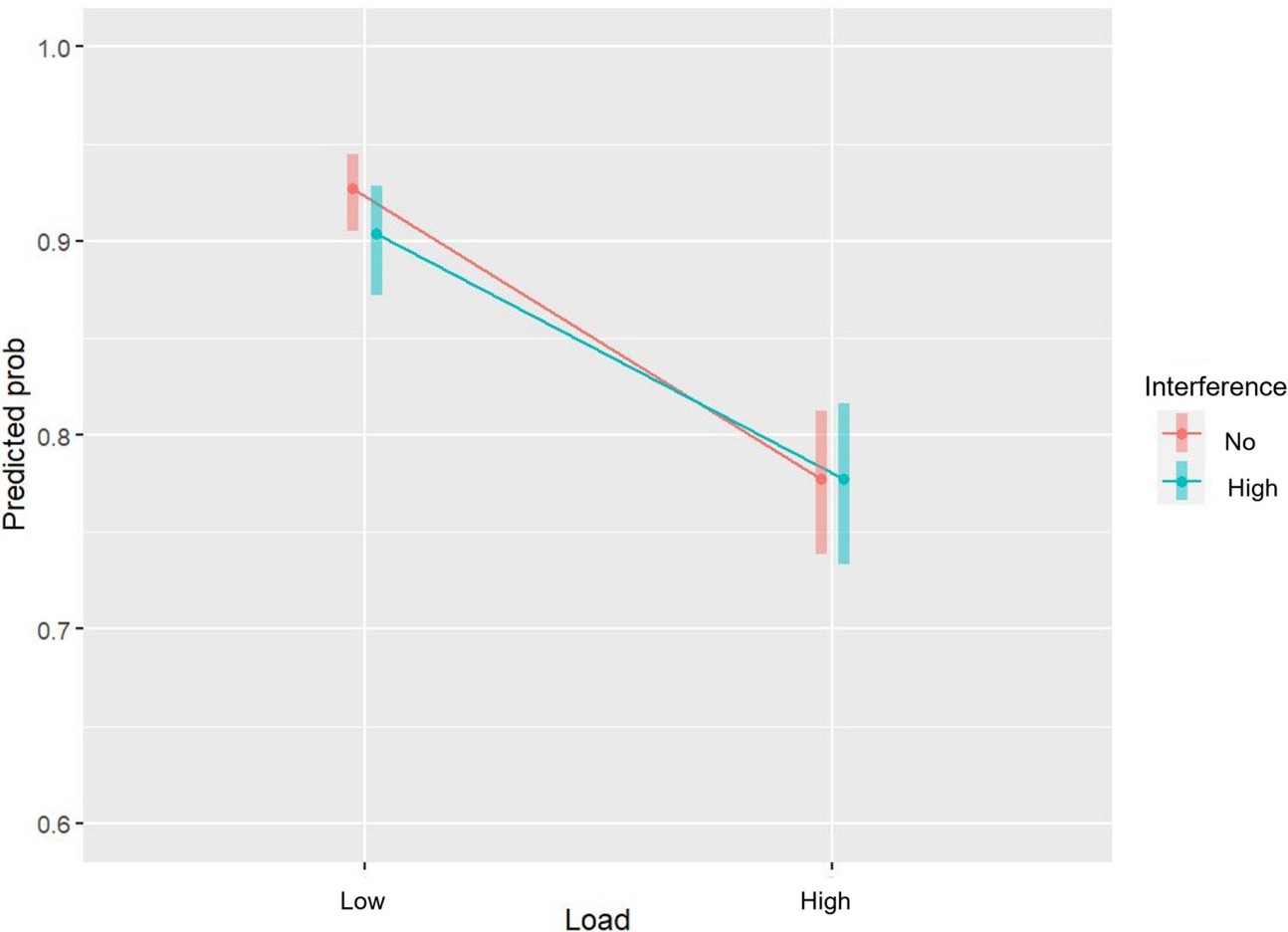

**Fig 3. Interaction plot for load and high interference.** The estimated probability of a correct responses (y-axis) for the high (turquoise graph) and no (red graph) interference conditions at low and high levels of working memory load (x-axis), based on the generalized linear mixed effects model. The error bars indicate the 95% confidence intervals of the estimates.

Since no proper power analysis was performed prior to the study, a sensitivity analysis based on post hoc power simulations was conducted. Simulations followed the guidelines from Kumle et al. [75], and we only considered the main manipulations of the experiment (working memory load and facial mimicry interference), which means that the simulation results can only be compared to the effects of those factors in the results from the generalized linear mixed effects model reported above. All effects but the interaction between high interference and working memory load (which was the critical test in the design) were set to a fixed value. The overall pattern of the simulation results suggested that beta weights larger than approximately .30 could be detected with at least 80% power, whereas beta weights in a lower range (.10-.20) revealed a power of less than 50% (for more details, see S1 Appendix). Thus, we cannot fully reject the possibility that poor sensitivity of our design might explain why we did not observe an effect of high interference when working memory load was high.

### Stimuli valence and arousal ratings

For arousal (Fig 4A), there was a significant main effect of the conditions happy ($\beta = 0.17$, $df = 11640$, $t = 50.02$, $p < .001$) and angry ($\beta = 0.17$, $df = 11640$, $t = 49.78$, $p < .001$). The same

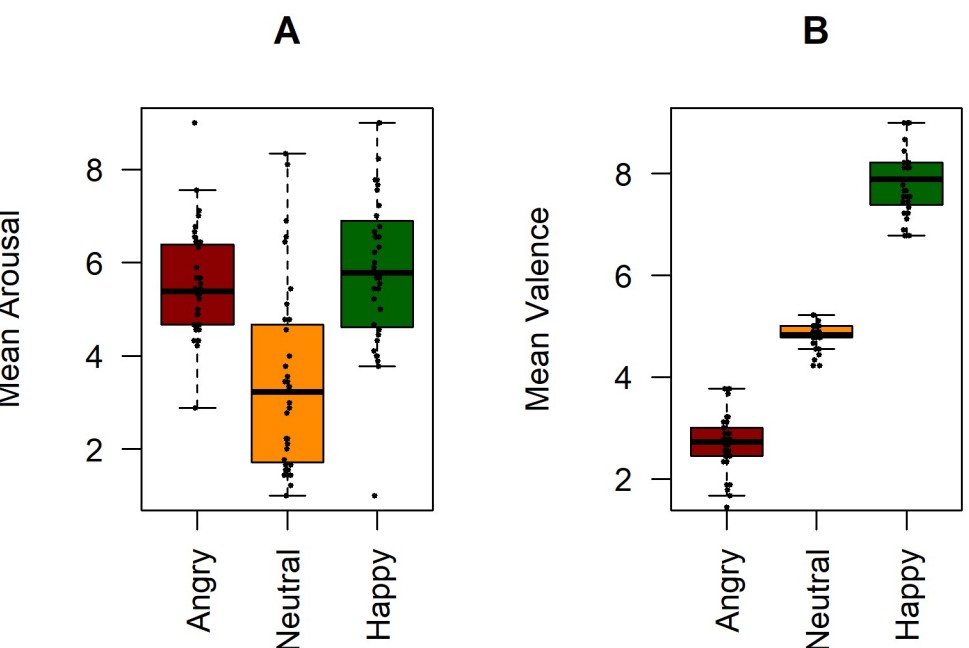

**Fig 4. Mean arousal and mean valence ratings as a function of emotion expression.** A) The average arousal rating (*y*-axis) displayed in bar plots for angry (red), neutral (yellow), and happy (green) emotion expressions (*x*-axis). The error bars indicate the 1.5 interquartile range, and the black dots are the individual means per participant. B) The average valence rating (*y*-axis) displayed as bar plots for angry (red), neutral (yellow), and happy (green) emotion expressions (*x*-axis). The error bars indicate the 1.5 interquartile range, and the black dots are the individual means per participant.

was visible for valence (Fig 4B) in conditions happy ($\beta$ = 2.30, *df* = 11660, *t* = 228.1, *p* < .001) and angry ($\beta$ = -1.37, *df* = 11660, *t* = -167.0, *p* < .001). Overall, pictures of angry expressions were rated as more negative compared to pictures of neutral and happy expressions, and more arousing than neutral expressions. Happy expressions were rated as more positive compared to pictures of neutral and angry expressions, and more arousing than neutral expressions. These results provide an approximate validation of the perceived emotional content of the pictures used in the experiment.

## Discussion

In the present work, we investigated the effect of facial mimicry interference on working memory for facial emotion expressions. Specifically, we tested whether a negative effect of behavioral facial mimicry interference on working memory precision increased with increasing working memory load. Working memory load had a strong negative effect on precision, but contrary to what was predicted, an effect of facial mimicry interference was only observed when working memory load was low. Thus, we found partial support for the notion that incongruent sensorimotor feedback impairs working memory for facial emotion expressions [35].

From a resource model perspective on working memory [5,6], load and precision of representations determine processing accuracy. We observed that working memory load had the expected detrimental effect on accuracy [1,7], corroborating earlier findings in the broader literature on the *n*-back tasks [1,76,77], and more specifically from the context of working memory for emotion expressions [12,52]. Further, precision was poorer when facial mimicry

interference was high, compared to when it was weak or absent, specifically when working memory load was low. Thus, as tentatively proposed in Baddeley's model of working memory for emotions [47–49], simulation of emotional content, as reflected by e.g. facial mimicry, might be a resource that improves the representational precision of emotion expressions in working memory, but perhaps only when load is low. Kuehne et al. [24] reported that invoked smiling enhanced short-term memory storage of happy emotion expressions, supporting the notion that simulation and sensorimotor feedback, as reflected in facial mimicry, influences not only perception but also brief memories of emotional input [35]. Our finding, that facial mimicry interference does not seem to influence precision when working memory load is high, might seem to contradict the finding of Kuehne et al. However, their experiment did not include an active manipulation of working memory load and was likely to induce only a weak working memory load. Thus, the results reported by Kuehne et al. might correspond to our finding at the low working memory load (1-back), and here we extend their work by showing that when working memory load is high (2-back), facial mimicry might play a limited role in successful performance. It should be noted that the facial mimicry manipulation applied by Kuehne et al. was intended to enhance a facial mimicry response, whereas here we wanted to interfere with facial mimicry. This might also contribute to the seemingly different results across studies.

To explain the unexpected finding that working memory load seems to suppress the negative effect of facial mimicry interference on precision, we cautiously propose that a general principle of the neurocognitive system is that when working memory demands increase, the system responds by filtering out potentially distracting information. The ability to suppress external stimuli—like background noise—is working memory dependent. Especially in high load or highly distracting conditions. This is well articulated in the task-engagement/distraction trade-off (TEDTOFF) model [78] and shown across different populations. Because the TEDTOFF model was primarily conceived and developed for audio-verbal stimuli, we only tentatively suggest that suppression and focal engagement of working memory resources in high-load conditions also apply to facial mimicry. However, we assume that the high interference condition produced incongruent information about the stimuli, and consequently, reduced the precision of the representation of the emotional expressions (i.e., internal interference). Thus, when processing becomes more difficult, the focal task might shield out potential distractors, and this could apply both when interference is external and when it is internal.

One previous study suggested that sensorimotor and visual information differentially contribute to the perception of facial emotion expressions [79]. This notion finds support in earlier models of face perception emphasizing the tracking of invariant visual features across individuals for successful identification of emotion expressions [80]. Situations in which a less costly, visual route for facial emotion processing could be utilized are likely characterized by higher cognitive demands, such as in larger group settings where it would be impossible to use sensorimotor simulation to process emotion expressions for everyone present [79]. As working memory demands increase, lexico-semantic representations and mechanisms subserving the online use of those representations become critical for the successful understanding of communicative signals [5] and other types of processing are down-prioritized [78,81]. The present study did not test whether access to semantic labels for facial input determines processing efficiency in demanding settings, and this possibility should be investigated in future studies. As a note related to this, neural responses during the blocking of facial mimicry indicate greater reliance on semantic retrieval [57], which links facial mimicry interference to active semantic processing. Further, one study reported that verbal but not visual working memory load reduced emotion recognition [21], suggesting that denied access to semantic labels reduces the precision of representations of emotion expressions. However, in a situation where the

observer only needs to process one face at a time, sensorimotor simulation might be useful both for representing the emotional state of the interlocutor [82] and to indicate an empathetic response or social relatedness [41–43].

Studies suggest that cognitive load might alter facial mimicry activity [44,45]. For example, Blocker and McIntosh [45] investigated facial mimicry responses, recorded using electromyography, to smiling and frowning faces that represented either individuals that the observer liked or individuals that were neutral to the observer. Recordings of mimicry responses were performed at both low and high working memory load, and when load was high, evidence of facial mimicry for smiling faces was only observed in response to facial expressions of individuals that the observer liked. Thus, working memory load suppressed mimicry responses when the motivation for social relatedness was low but not as much when motivation was high. It might be the case, that facial mimicry is suppressed by cognitive load of any kind, but that contextual factors might in turn surpass this inhibiting effect [42,43,83]. Blocker and McIntosh finding, that increased working memory load suppress facial mimicry responses, suggests that facial mimicry might not play a crucial role in accurately representing facial emotion expressions when working memory load is high. However, in the present study, we did not see that facial mimicry interference influenced precision when working memory load was high. It should be acknowledged that this surprising result could be due to some characteristics of the present sample or features of the task design, possibly limiting the precision of the statistical model even though the total number of observations per condition was high (36 participants, and 54 trials per load by mimicry interference condition). Thus, the effect of facial mimicry when working memory load is high should be investigated in larger studies using different experimental settings before any firm conclusions can be drawn.

In one model of working memory for emotions [47–49], inputs are assumed to be evaluated on a positive-negative internal valence scale inducing a feeling state simulation. Although this is related to facial mimicry, as sensorimotor simulation could activate feeling states [35], it is not likely to be determined by facial mimicry alone [39]. Thus, we cannot say that participants did not at all use simulation to solve the task, since simulation of feeling states might be driven by mechanisms not targeted with the type of manipulations applied in the present study [84]. Unfortunately, we did not assess participants' feeling states during the experiment, nor which strategies they used to solve the task. However, based on the valence and intensity ratings that participants completed, the stimuli seemed to invoke the sought emotional dimensions along the positive-negative continuum. This provides some evidence that the different categories of emotion expressions invoked different feelings in the participants. That the valence and arousal ratings were associated with task performance, further suggests that the perceived emotional content of the stimuli might explain some variability in the precision of working memory representations of emotion expressions. The evidence is somewhat mixed and heterogeneous [85], but some previous studies suggest that the emotional valence of the stimuli might have an impact on working memory performance [86–88], and specifically, working memory for facial emotion expressions [12,13]. In addition, it has been reported that positive emotions are stored with higher precision in working memory than negative emotions [89], a finding that we replicated here. Future studies investigating the effects of facial mimicry and working memory load should consider potential interactions with the perceived valence and arousal dimensions of the stimuli.

One limitation in this and most of the previous studies investigating the role of facial mimicry in the identification and processing of facial emotion expression, is that little is known about how the behavioral mimicry manipulations applied interfere with mimicry activity. When an effect is observed on behavioral precision, it is typically assumed that there also was an effect on mimicry activity, without this being directly observed. Facial mimicry is measured

with electromyography (EMG) [37], a method that might not be compatible with most mimicry interference manipulations since the method is based on the placement of electrodes to the face and sensitive to disturbances. The present study did not include a validation of whether the facial mimicry manipulations that we used induced the sought effects by applying EMG measurements. Instead, we assumed that the manipulations were valid based on observed changes in behavioral performance reported in previous studies [31,66]. In addition to EMG, it might be possible to test the validity of facial mimicry manipulations by coding filmed emotion expressions when manipulations are active, using the Facial Action Coding System [90]. Whether and how facial mimicry manipulations interfere with mimicry activity should be considered carefully in future studies and validated using either EMG or video coding. A final and potentially major limitation was the lack of non-face control stimuli. With this control, we could have tested whether the interference effect that we observed for the low-load condition is, as would be expected, specific for facial input. At the same time, it was only the high interference and not the low interference condition that impaired precision, which speaks for that the observed pattern of results was not driven by a general interference effect. However, had the design included a control condition where the task was to store non-face stimuli, we would feel more confident with our conclusion.

## Conclusions

Facial mimicry might influence the precision of representations of facial emotion expressions when load on working memory is low. Thus, sensorimotor feedback represents a useful source of information for the processing of emotion expressions when the conditions allow for it. Everyday life involves effortful processing of emotion, and an impaired ability to recognize and make use of emotion expressions might fall short due to impaired precision of such expressions in working memory.

## Supporting information

**S1 Appendix. Post hoc power simulations.**
(DOCX)

## Acknowledgments

We would like to thank Erik Olsson for help with the data collection.

## Author Contributions

**Conceptualization:** Emil Holmer, Jerker Rönnberg, Erkin Asutay, Mattias Ekberg.

**Data curation:** Emil Holmer, Carlos Tirado, Mattias Ekberg.

**Formal analysis:** Emil Holmer, Carlos Tirado.

**Funding acquisition:** Jerker Rönnberg.

**Investigation:** Mattias Ekberg.

**Methodology:** Emil Holmer, Jerker Rönnberg, Erkin Asutay, Mattias Ekberg.

**Project administration:** Emil Holmer.

**Resources:** Emil Holmer, Jerker Rönnberg.

**Software:** Emil Holmer, Carlos Tirado, Mattias Ekberg.

**Supervision:** Emil Holmer, Jerker Rönnberg.

**Validation:** Carlos Tirado.

**Visualization:** Carlos Tirado.

**Writing – original draft:** Emil Holmer, Carlos Tirado, Mattias Ekberg.

**Writing – review & editing:** Emil Holmer, Jerker Rönnberg, Erkin Asutay, Carlos Tirado, Mattias Ekberg.

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
