## [Decision Letter · Decision Letter 0]

15 Aug 2023

PONE-D-23-06734Facial mimicry does not influence memory precision of emotion expressions when working memory load is highPLOS ONE

Dear Dr. Holmer,

Thank you for submitting your manuscript to PLOS ONE. After careful consideration, we feel that it has merit but does not fully meet PLOS ONE’s publication criteria as it currently stands. Therefore, we invite you to submit a revised version of the manuscript that addresses the points raised during the review process.

I want to begin by apologizing for the lengthy delay in getting a recommendation to you. I had sustained difficulty securing reviewers for your submission, with some reviewers failing to submit their report, requiring me to secure new reviewers and further delaying a final recommendation. None of this affected my judgement of your manuscript, which I personally enjoyed reading, it's just an unfortunate pattern editor have been dealing with since COVID.

I was fortunate to secure two expert reviewers with knowledge of the field, each of whom provided insightful and helpful reports. As you will see, both reviewers and I had a positive opinion on the quality and clarity of the manuscript. Overall, the manuscript is well written, appropriately cited, and clearly identifies a gap in the mimicry literature; I personally enjoyed reading your work. However, both reviewers and I took issue with certain aspects of the narrative and analyses.

There was consensus across reviewers, including myself, that facial mimicry was not the observed variable in the study, and that this needs to be made much clearer to the reader. The narrative makes mimicry a central piece of the story, yet mimicry was not measured (with EMG or video motion); rather, it was facial expression recall accuracy that was measured. Like R2, the results of the interaction are interesting, and it would be helpful to contextualize these findings with typical n-back literature findings. R1 makes the insightful comment that the study had no control condition with no-facial stimuli; a discussion here is warranted. Reporting of your mixed model can also be improved for clarity. Finally, interpretation of the null, as indicated by R2 is a serious point for consideration given the lack of a prior sample size estimate. Below you will find comments of the two reviewers, along with own additional suggestions.

Alexthymia is considered a personality trait, not a medical condition or disorder. See Ricciardi et al, 2015 (Neuropsychiatry) for reference material, or you can just drop the reference to medical condition.Your manuscript is missing a strong justification for sample size. I won’t ask for a posthoc sample size estimate, as this is meaningless. Rather, see Lakens, 2022 for sample size/power justification and follow their advice that matches your situation. In future, please consider the use of simR as a suitable sample size estimate in mixed models. https://humburg.github.io/Power-Analysis/simr_power_analysis.htmlYour mixed model reporting is missing many key details. Did you investigate the assumptions of mixed models? See Harrison et al, 2018. Normality of residuals, homogeneity of variances etc? sjPlot provides straightforward visual diagnostics.What form of p-value correction are you reporting, Satterthwaite or Kenward-Roger? Luke 2017 suggests that KR is the gold standard. Either way, please include in your manuscript.What was your random effects structure? I note in your script that with (1|fp), you’re not exploring by-participant random slopes for load or interference; yet a maximal RE structure is recommended to avoid Type I errors; Barr et al, 2013.Ideally, you would report other aspects of your model (optimiser used, number of iterations). Specify if you optimised with REML or ML.Please specify in the manuscript the distribution used (logit)I agree with R1’s concerns regarding interpretation of the interaction. You may be better positioned to code the contrasts yourself. You may find this tutorial helpful - https://marissabarlaz.github.io/portfolio/contrastcoding/. Optionally, you may want to consider use of the emmeans package and performing pairwise comparisons.Fig 2 axes need better labelling. It is not clear from the Y-axis or caption what your DV is. Please state it clearly (raw accuracy, I believe, from the letter matching n-back task).Rating of Radbound images was a good addition, though not really needed given they are validated stimuli. I am curious why you did not tie your AV data to your accuracy data, as also suggested by R2. That is, did you consider adding A/V as continuous predictors in your glmer model?

We look forward to receiving your revised manuscript.

Kind regards,

Steven R Livingstone

Academic Editor

PLOS ONE

Journal Requirements:

2. We note that Figure 1 includes an image of a participant in the study. 

Reviewers' comments:

Reviewer's Responses to Questions

**Comments to the Author**

1. Is the manuscript technically sound, and do the data support the conclusions?

Reviewer #1: Yes

Reviewer #2: Partly

2. Has the statistical analysis been performed appropriately and rigorously? 

Reviewer #1: Yes

Reviewer #2: No

3. Have the authors made all data underlying the findings in their manuscript fully available?

Reviewer #1: Yes

Reviewer #2: No

4. Is the manuscript presented in an intelligible fashion and written in standard English?

Reviewer #1: Yes

Reviewer #2: Yes

5. Review Comments to the Author

Reviewer #1: This experiment examines the role of facial mimicry in working memory for images of facial expressions. Overall, the methods are nice and straightforward, although the lack of non-face stimuli is a definite limitation. I have a few suggestions for improving the regression model specification and interpretation. I also find myself disagreeing about the authors’ takeaway, that “Facial mimicry does not play a critical role in working memory for emotion expressions” (p. 21; see my point #6).

METHOD

1. A non-trivial limitation of the study design is a lack of non-face stimuli. We cannot know whether the effects of facial mimicry interference are specific to facial expression working memory, or whether the manipulation was just generally distracting and would have impacted accuracy for any type of stimuli. This needs to be addressed in the Discussion.

ANALYSES AND RESULTS

2. I think the dependent variable in the regression is better described as “whether the response on a given trial was correct or incorrect” or something like that to make it explicit that it’s at the trial level and dichotomous. The current phrasing, “the probability of a correct response as the dependent variable” (p. 11), makes it sound like you’ve converted raw trial-level data into percentages.

3. The Load variable is a numeric variable coded as 1 and 2—with the interaction term, this means you’re interpreting the main effects for the Interference dummy variables at a nonexistent level of Load, 0 (0-back trial?). If you relevel Load so 1-back is 0 and 2-back is 1, then you can interpret your Interference main effects as being the effect of facial mimicry interference for the 1-back trials. Here’s a blog post about how changing what 0 means for one variable changes the interpretation of another variable when there’s an interaction term: https://www.r-bloggers.com/2018/05/regression-with-interaction-terms-how-centering-predictors-influences-main-effects/

4. It’s really tricky to describe interactions with dummy-coded variables, and I’m not sure that this sentence (p. 12) is complete: “The most important finding of our study was an interaction between working memory load and low interference (β = -.266, df = 34, p < .044) and load with no interference (β = -.270, df =34, p < .042).” It’s really 1) the interaction between Load and the dummy variable comparing low and high interference and 2) the interaction between Load and the dummy variable comparing no interference and high interference. Note that the two interference variables are using “high interference” as the reference level and comparing “low” and “none” to it. Be similarly careful with your wording when explaining the main effects for interference.

5. The labels in Figure 4 are unclear to me. Based on the in-text description, you’ve transformed the model coefficients into probabilities (which I tried doing myself and got different values, so maybe showing the formula would help). I would change the labels. For instance, make it clearer that the probability for “Load” is how much the probability of a correct response changes from low to high load. And the “No” probability is how much the probability of a correct response changes from the “High interference” to the “no interference” conditions. Honestly, I’m not sure if that figure is even worth using since it’s pretty confusing and somewhat redundant with Figure 3. Do we really care about the relative effect sizes of these different variables?

DISCUSSION

6. It’s interesting that you found facial mimicry impairment reduced accuracy on the 1-back trials but not the 2-back trials. One possible interpretation is facial mimicry helps you keep the most recent face in working memory, but the 2-back face’s associated simulation is overridden by the 1-back face. It’s hard to imagine how sensorimotor activity could maintain representations of multiple facial expressions at once when they involve opposing facial actions. This interpretation does not lead me to the conclusion that you “did not find convincing support for the notion incongruent sensorimotor feedback impairs working memory for facial emotion expressions although it might impair precision in the absence of a high working memory load.” (p. 15). As an analogy, let’s say I’m trying to keep a melodic phrase in my working memory by repeating it in my head (which recruits auditory system). If I am unable to do it for two melodies simultaneously, you wouldn’t then conclude that the auditory system wasn’t playing a role—you’d conclude that my auditory working memory can only handle 1 melody at a time. I’m happy to be convinced otherwise if the authors have already ruled out this interpretation for themselves.

7. I’m confused by this interpretation in the discussion section (p. 17): “working memory demands increase the neurocognitive system responds by filtering out potentially distracting information, as proposed in the task-engagement/distraction trade-off (TEDTOFF) model.” How is sensorimotor activity distracting information? Isn’t it (theoretically) aiding task performance? Distracting externally-generated auditory input in the Sörqvist et al. study is different from internally-generated and task-relevant facial mimicry.

8. The discussion is a tad repetitive. For instance, the paragraph starting on p. 19 with the phrase “In addition to the potential role of mimicry” was a restatement of earlier ideas.

9. “…and the sample size was too small to test a three-way interaction” (p. 21). Is this true, given that the additional variable is also within-subject?

GENERAL

10. The paper needs to be proofread, as I encountered the occasional typo or grammar issue throughout. For instance, this sentence on p. 18 is not grammatically correct and is difficult to follow: "Blocker and McIntosh finding of suppressed facial mimicry responses when working memory load increases, speaks for that facial mimicry might not play a crucial role in accurately representing facial emotion expressions when working memory load is high.” Possible rewording: “Blocker and McIntosh’s finding, that increases in working memory load suppress facial mimicry, suggests that facial mimicry might not play a crucial role in accurately representing facial emotion expressions when working memory load is high.”

Reviewer #2: In this manuscript, the authors systematically address a gap in the literature regarding the effects of interfering with facial mimicry of emotions on working-memory performance at different loads. For this purpose, they assessed 36 students in their 20s (72% female) on a 2 x 3 within-subjects design with two levels of working memory and three levels of interference with facial mimicry. The main DV was accuracy on an emotional n-back task. The Introduction is well-written, presenting relevant background to support a clear rationale. The authors expected a cumulative effect of the 2 factors on accuracy, but the observed interaction was such that an interference effect was only significant in the low working-memory load condition, where low and no interference appeared similarly superior to high interference. A main effect of load was also observed. These effects are generally consistent with prior studies. What is most novel is the lack of interference effects in the high working-memory load condition. Although EMG and subjective reports were not included to verify the interference manipulations, the authors adequately address these limitations. Overall, they conclude that facial mimicry does not play a critical role in working memory for emotional expressions, but may influence performance when working-memory demands are low. However, the interpretive claims rest on the null finding at high load, which deserves further attention, along with other aspects regarding the design and analyses.

MAIN COMMENTS

1. As above, the main conclusion that facial mimicry does not play a critical role in working memory for emotional expressions rests on the failure to observe an effect of interference in the 2-back condition. The Discussion briefly notes that performance was still above chance (p.18) and the figures suggest there was still substantial variation at high load. What was chance? Was the distribution skewed? A counter-argument should be made to the alternative explanation that the absence of effect may be due to a floor effect or restricted range. Moreover, Bayes factors would be helpful towards supporting the strong claims in favour of the null; at present, null hypothesis significant testing cannot support the claims as currently phrased.

2. The basis for the sample size should be elaborated or clarified. The methods indicate it was “determined based on previous studies…and thirty-six students participated” (p.7). Does this mean that N=36 was somehow optimal or minimal within a range observed in prior studies? Or did the authors obtain an effect size to calculate and estimated sample size a priori. If so, for what effect(s) of interest? Even if sufficiently powered, it is a relatively small sample size from a reliability / replication standpoint. The Discussion also notes that the “sample size was too small to test a three-way interaction” including valence (p.21). Why wasn’t this considered a priori given the depth of consideration paid to valence in the paper and the field? Additionally, as valence would be embedded within the multilevel model, I’m also not convinced the current design lacks sufficient power.

3. While the basic script is posted on OSF, the analytic models could be further elaborated. For instance, how were the levels of load and interference modeled and coded (dummy variables)? This is important for interpreting the sign of the regression coefficients. Why did the high interference condition serve as the reference category in these analyses? Why not the no-interference condition to allow for potentially more meaningful assessments of the interactions between low and high interference with load (vs none and low)? Why was a mixed-effect model used to assess n-back performance, but a traditional ANOVA used to assess valence and arousal/intensity ratings?

4. Moreover, a fair bit of attention is devoted to validating the valence/arousal ratings. The Discussion notes the role of context (p.18) and potential moderation by expression type (p.21). As above, why not also include it as a factor of interest in the analyses? This may prove important if variation across expression types are masking an effect of interference when averaged across valence.

5. The figures suggest sizable variation across participants in accuracy, particularly under high load (although it’s not clear what the error bars reflect or the underlying distribution; see below). It may be useful to account for some of this variability in the analysis.

a. For example, although counterbalanced, were there any order effects of the load or interference conditions? The same stimuli repeated across trials. Namely, 9 face identities were each presented with the 3 emotional expressions (27 images total), such that across the 6 blocks, each image was presented twice and each facial identity 6 times. Might this have introduced some habituation or mnemonic interference across trials?

b. Sources of individual differences may also be contributing to error variance. A practice task was used to reduce baseline performance variability. However, it may be useful to harness the verbal working-memory scores (LNS subscale) as covariate if it relates to task performance(?); consider the arguments made for verbal working-memory involvement particularly at high loads. Likewise, what about other participant characteristics? For example, all the stimuli were of women; did participant sex moderate the results?

MINOR POINTS

6. Alexithymia is not typically considered a “medical condition” per se (p.4). Although there are different views on the construct, it can present on a subclinical spectrum and some view it as a personality trait, for example.

7. Did the data meet the assumptions for the main analyses (i.e., normality, homogeneity of variance, no outliers)? Only the data for the valence/arousal ratings are provided on OSF.

8. Related, while the valence/arousal figures are box plots, the main n-back accuracy findings are displayed as predicted probabilities with some margin of error (bars not defined). The distributions are thus unclear – the authors might consider displaying a dot plot or at least a box plot for these findings as well.

9. I don’t see any captions for the figures in the file for review – are they missing?

6. PLOS authors have the option to publish the peer review history of their article (what does this mean?). If published, this will include your full peer review and any attached files.

Reviewer #1: No

Reviewer #2: No

---

## [Author Response · Author response to Decision Letter 0]

20 Dec 2023

The responses are included in the "Response to Reviewers" document. I also included the responses below. However, I could not insert figures, and the formatting of the tables were awful. Therefore, I refer to the "Response to reviewers" document to access the figures/tables.

Editor comments:

I want to begin by apologizing for the lengthy delay in getting a recommendation to you. I had sustained difficulty securing reviewers for your submission, with some reviewers failing to submit their report, requiring me to secure new reviewers and further delaying a final recommendation. None of this affected my judgement of your manuscript, which I personally enjoyed reading, it's just an unfortunate pattern editor have been dealing with since COVID.

I was fortunate to secure two expert reviewers with knowledge of the field, each of whom provided insightful and helpful reports. As you will see, both reviewers and I had a positive opinion on the quality and clarity of the manuscript. Overall, the manuscript is well written, appropriately cited, and clearly identifies a gap in the mimicry literature; I personally enjoyed reading your work. However, both reviewers and I took issue with certain aspects of the narrative and analyses.

There was consensus across reviewers, including myself, that facial mimicry was not the observed variable in the study, and that this needs to be made much clearer to the reader. The narrative makes mimicry a central piece of the story, yet mimicry was not measured (with EMG or video motion); rather, it was facial expression recall accuracy that was measured. Like R2, the results of the interaction are interesting, and it would be helpful to contextualize these findings with typical n-back literature findings. R1 makes the insightful comment that the study had no control condition with no-facial stimuli; a discussion here is warranted. Reporting of your mixed model can also be improved for clarity. Finally, interpretation of the null, as indicated by R2 is a serious point for consideration given the lack of a prior sample size estimate. Below you will find comments of the two reviewers, along with own additional suggestions.

R: Regarding the delay, we are aware of the issues of finding reviewers, and appreciate your effort in doing this. You also seem to have been successful, the comments from both reviewers were useful. We are very grateful for getting the opportunity to submit a revised version of this manuscript. Also, we should apologize for taking perhaps a bit too much time for our revision. From what we understand, you seem to be interested in what we do here, and we wanted to make sure that we did not miss any part that needed to be changed.

The comments in the previous round of review included several different aspects related to the general narrative of the paper, some selections made in the design and analyses, and our conclusions. Regarding the narrative, we now place the study more clearly in the context of the WM literature, arguing that facial mimicry might contribute to the precision of representations based on a resource model perspective of WM. We have also tried to clarify that we did behavioral manipulations of facial mimicry, and that the outcome we were interested in investigating was WM recall for emotion expressions. These changes have mostly impacted on the introduction and the discussion sections. We have also updated the rationale for the sample size, based on Laken’s terminology, and the description of the analysis, after changing the analysis itself based on the detailed comments below. The most important changes are outlined here: 

- The title was changed to “Facial mimicry interference reduces working memory accuracy for facial emotion expressions when task load is low but not when it is high” which acknowledges that we actually do see an effect when load is low, and that we did not find evidence of the same effect when working memory load is higher. This is different from stating that facial mimicry does not influence working memory processing.

- In the introduction, we now start from the perspective of WM, and ask if (behavioral) interference of (presumed) facial mimicry will produce a negative effect on WM precision. Our perspective on WM is based on a resource model which emphasizes the interacting effects of storage demands and precisions of representations in WM processing. With our manipulations, we assume that facial mimicry interference will reduce precision (probably through impaired sensorimotor simulation), and the increase in load will lead to increased storage demands. Based on this, the argument is that load and interference will produce interacting negative effects on WM recall. However, since we only see the effect of interference when load is low, we think that our data suggest that facial mimicry influences precision when WM storage demands are not too high. This is also reflected in the discussion of the results, and the conclusions. We think this is in line with what you suggest, but expressed within the WM framework we usually apply.

- In the first paragraph of the Discussion (p. 17), we conclude: “Working memory load had a strong negative effect on precision, but contrary to what was predicted, an effect of facial mimicry interference was only observed when working memory load was low. Thus, we found partial support for the notion that incongruent sensorimotor feedback impairs working memory for facial emotion expressions [35].” As in the update title, here we both say that there is an effect of facial mimicry when load is low, and that we did not find evidence of the same effect when working memory load was higher. The same idea is also expressed in the updated conclusions (p. 22): “Facial mimicry might influence the precision of representations of facial emotion expressions when load on working memory is low but perhaps not when load is high. Thus, sensorimotor feedback represents a useful source of information for the processing of emotion expressions when the conditions allow for it.”

- We have added a rationale for the sample size based on Laken’s terminology (p. 8): “The sample size was determined based on resource limitations and heuristics [56]. More specifically, previous studies with similar manipulations and designs showed statistically significant effects with samples in the range of 20 to 50 participants [22,25,29,30,57] and such an approach was also deemed feasible in our study given the available funding and the planned balancing of conditions. Thirty-six participants were recruited, and our expected sample size was reached.”

- We have updated the description of the analysis with more details (p. 12-14): “To investigate our main prediction that a negative effect of facial mimicry on emotion expression precision increases with increasing working memory load, we performed a generalized linear mixed effects model with the within-group factors specified as working memory load (two levels: low, 1-back, and high, 2-back) and facial mimicry interference (three levels: no, low, and high), as well as their interaction. The dependent variable was whether the response on a given trial was correct. Control factors included the block order to account for potential habituation and mnemonic effects, as well as individual mean valence and mean arousal estimates for the three types of emotional expressions in the stimuli material (angry, happy, and neutral) to model individual differences along these dimensions. The random effect structure included correlated intercepts and slopes for load at the level of the individual. Analysis was performed in R statistical software [69] using the glmer function from the lme4 package [70] for model estimation. For testing the simple effects of interactions, the emmeans package [71] was used. To deal with the binary outcome variable, a logit-link function was applied. The model was estimated using maximum likelihood estimation with the bobyqa optimizer and 1000 iterations. Satterthwaite approximation of degrees of freedom was applied to test fixed effects. Working memory load was dummy-coded, using one variable with low load as the reference at 0. Facial mimicry interference was also dummy-coded, using two variables, one for low interference and another for high interference, with no interference as the reference at 0 in both. We used the package sjPlot [72] to run comprehensive regression diagnostics for our main model, including assessments for variance, homoscedasticity, normality, random effects, and outliers, which consistently validated the model’s conformity to the assumptions. To test the validity of the emotion expressions in the pictures, mean valence and mean arousal ratings for each of the categories angry, neutral, and happy expressions were compared, in two separate analyses. The random effect structure in these analyses included random intercepts at the level of the individual. A more complex random effect structure created convergence issues. The assumption of normality was violated for the rating data since it showed clustering at several areas of the curve. Hence, a Gaussian Mixture Model (GMM) approach was used for these analyses. This approach was applied by using the package mixtools [73] and the function normalmixEM to generate the best fit. It allowed the model to capture complex patterns in the data (non-linear) and perform with accuracy despite the assumption of normality being violated. The effects of angry and happy emotion expressions were modelled using two dummy-variables with neutral emotion expressions as the reference at 0. The model was estimated using maximum likelihood estimation with the Nelder-Mead optimizer and 1000 iterations. No data was missing, and the significance level was set to α = .05 in all statistical analyses. The data file (.csv) and analysis script (in R) for this study are available on the Open Science Framework (see: https://osf.io/yhq47/?view_only=9d22e7feb6e043829c1735bc191fbc44).”

- We have changed the main analysis, using dummy-coded variables instead of factor variables, and by adding several covariates. The new results reads (p. 14-15): “For accuracy, there was a main effect of working memory load (β = -1.30, df = 35, p < .001), which means that the higher the load the lower the performance accuracy (see Table 1). Further, there was a negative main effect of high interference, β = -0.61, df = 35, p = .009, but no main effect of low interference, suggesting that performance accuracy in the high interference condition only was poorer than in the no interference condition. We found positive main effects of valence (β = 0.078, df = 35, p < .001) and arousal (β = 0.26, df =35, p = .003), which means that performance accuracy improved when the mean valence and mean arousal were higher. Finally, the main effect of block order was also positive and statistically significant (β = .076, df = 35, p < .001), therefore, participants improved as they progressed in the task. However, the most important finding of our study was that the interaction between working memory load and high interference was statistically significant (β = 0.30, df = 35, p = .024, displayed in Fig 3), but the interaction between working memory load and low interference was not (see Table 1). Following up the simple main effects on the statistically significant interaction revealed that when working memory load was high (2-back), the effect of high interference was not significant (β = 0.00, df = 35, p = 1.00). The pattern and direction of the main and interaction effects suggest that there was a negative effect of the high interference condition when working memory load was low, but not when the load was high (see Fig 3).”

For responses to all specific comments, see details below. We have attached both a clean version, and a version with tracked changes. The tracked-changes version looks terrible (because of the number of changes), but from what we understood, we should include both in the re-submission. Let us know if we need to prepare another version.

 

Comments from editor

R: We think that the previous version had the wrong formatting of the figures and the table, as well as some minor formatting issues in the text. We apologize for this, and in the new version we have changed the formatting to align with the style requirements. 

2. We note that Figure 1 includes an image of a participant in the study. 

R: This is changed in the Fig 1 legend (p. 10) and added in adjunct to the ethics statement (p. 8).

• Alexthymia is considered a personality trait, not a medical condition or disorder. See Ricciardi et al, 2015 (Neuropsychiatry) for reference material, or you can just drop the reference to medical condition.

R: Thanks for pointing this out. It now reads “which is the impaired ability to represent, recognize, and verbally label emotional states” (p. 4). 

• Your manuscript is missing a strong justification for sample size. I won’t ask for a posthoc sample size estimate, as this is meaningless. Rather, see Lakens, 2022 for sample size/power justification and follow their advice that matches your situation. In future, please consider the use of simR as a suitable sample size estimate in mixed models. https://humburg.github.io/Power-Analysis/simr_power_analysis.html

R: Thanks for the suggestion on using simR for future studies. We have read the Lakens (2022) paper, and the situation best describing the justification for our sample size is resource limitations combined with heuristics. Data collection was based on what we saw as a reasonable number of participants in relation to previous studies (reporting significant results) and the available funding. We also naïvely assumed that the effects would probably be large, since that is usually what we observe in the working memory studies we do (manipulations with effects around d = 1, or stronger, are common). We have updated the justification for the sample size, using Lakens (2022) as a reference (see p. 8):

“The sample size was determined based on resource limitations and heuristics [56]. More specifically, previous studies with similar manipulations and designs showed statistically significant effects with samples in the range of 20 to 50 participants [22,25,29,30,57] and such an approach was also deemed feasible in our study given the available funding and the planned balancing of conditions. Thirty-six participants were recruited, and our expected sample size was reached.”

• Your mixed model reporting is missing many key details. Did you investigate the assumptions of mixed models? See Harrison et al, 2018. Normality of residuals, homogeneity of variances etc? sjPlot provides straightforward visual diagnostics.

R: Thanks for pointing this out. We now comment of the assumption checks of the model(s) in the paper. All assumptions were tested using

---

## [Decision Letter · Decision Letter 1]

13 Mar 2024

PONE-D-23-06734R1Facial mimicry interference reduces working memory accuracy for facial emotion expressions when task load is low but not when it is highPLOS ONE

Dear Dr. Holmer,

Thank you for submitting your manuscript to PLOS ONE. After careful consideration, we feel that it has merit but does not fully meet PLOS ONE’s publication criteria as it currently stands. Therefore, we invite you to submit a revised version of the manuscript that addresses the points raised during the review process.

 Thank you for your patience while we collected the reviewer's feedback. As one of the original reviewers was unable to accept the revision task, a third reviewer (R3) was brought on to provide a new review. The original Reviewer 1 was happy with your revisions and is ready to accept the manuscript. However, R3 has asked for revisions. Most of these relate to statistical reporting. While they do not require the collection of new data, they are significant enough to be classified as major revisions. Therefore, I invite you to revise and resubmit the manuscript to address R3's concerns.

We look forward to receiving your revised manuscript.

Kind regards,

Steven R Livingstone

Academic Editor

PLOS ONE

Reviewers' comments:

Reviewer's Responses to Questions

**Comments to the Author**

1. If the authors have adequately addressed your comments raised in a previous round of review and you feel that this manuscript is now acceptable for publication, you may indicate that here to bypass the “Comments to the Author” section, enter your conflict of interest statement in the “Confidential to Editor” section, and submit your "Accept" recommendation.

Reviewer #2: All comments have been addressed

Reviewer #3: (No Response)

2. Is the manuscript technically sound, and do the data support the conclusions?

Reviewer #2: Yes

Reviewer #3: Partly

3. Has the statistical analysis been performed appropriately and rigorously? 

Reviewer #2: Yes

Reviewer #3: Yes

4. Have the authors made all data underlying the findings in their manuscript fully available?

Reviewer #2: Yes

Reviewer #3: Yes

5. Is the manuscript presented in an intelligible fashion and written in standard English?

Reviewer #2: Yes

Reviewer #3: Yes

6. Review Comments to the Author

Reviewer #2: (No Response)

Reviewer #3: In the revised manuscript titled “Facial mimicry interference reduces working memory accuracy for facial emotion expressions when task load is low but not when it is high,” the authors investigated the effect of blocking spontaneous facial mimicry on working memory recall for facial emotion expressions. They reported that high-level mimicry interference was effective for 1-back but not for 2-back recalls. The authors made great efforts in addressing comments from the editor and reviewers 1 & 2. However, key points about the study’s affordance to provide enough precision for their observed effects were not addressed.

1. The authors stated that they have followed Lakens’ suggestions (Lakens, 2022) in sample size justification. While Lakens provided six principles, the authors only stated reasons for resource limitations and heuristics. I want to emphasize that the authors need to demonstrate whether their present study provided precision in effect estimation for what they expected to observe and what they actually observed. The issue of precision in effect estimation is demonstrated in (Lakens & Evers, 2014) Table 1. Small effects require a large sample size to reach the point of stability in effect estimation. With an insufficient sample size, the observed effects are likely unreliable and unlikely to be reproduced. Therefore, I would like to request explicitly:

a. A sensitivity analysis to determine, at the current design and sample size, what the minimally detectable effect size (MDES) is with 80% power. Note that this has nothing to do with your observed effect size. In Lakens’ terms (Lakens, 2022), this is “which minimal effect size will be statistically significant.”

b. Reports of observed effect sizes for all statistical results (Green & MacLeod, 2016; Nakagawa & Schielzeth, 2013).

c. Considering whether the present study afforded enough precision in effect estimation for the significant and null results they reported in the manuscript. Suppose the effect size reported is much smaller than the present study's precision. In that case, the results reported should not be considered conclusive, and I will have difficulty recommending publishing this study as it is.

2. The discussions focused on the unexpected results and the limitations. While it is okay to speculate that lexico-semantic representations might come into play when working memory demands increase (lines 415-434), the authors should note explicitly that the present study provided no evidence of whether it was the case with their participants.

3. The lack of facial EMG validation of the mimicry interference was apparent. The authors first defended that the placement of electrodes is not compatible with the interference manipulation. Still, they later cited (Davis et al., 2017) who did such validation to strengthen their “assumption” that their manipulation was successful. I suggest it is enough to acknowledge that some validation should have been done, but it is missing in the present study.

4. As mentioned by the editor, participants’ facial video recordings might also help validate the manipulation. While the automated FACS (e.g., OpenFace or Py-Feat) might be ineffective in estimating AU12 and AU4 with the plastic foam rob and tapes in place, human FACS raters might be able to tackle this issue. Thus, facial video recordings could be mentioned as a validation method.

Bibliography

Davis, J. D., Winkielman, P., & Coulson, S. (2017). Sensorimotor simulation and emotion processing: Impairing facial action increases semantic retrieval demands. Cognitive, Affective, & Behavioral Neuroscience, 17(3), 652–664. https://doi.org/10.3758/s13415-017-0503-2

Green, P., & MacLeod, C. J. (2016). SIMR: An R package for power analysis of generalized linear mixed models by simulation. Methods in Ecology and Evolution, 7(4), 493–498. https://doi.org/10.1111/2041-210X.12504

Lakens, D. (2022). Sample Size Justification. Collabra: Psychology, 8(1), 33267. https://doi.org/10.1525/collabra.33267

Lakens, D., & Evers, E. R. K. (2014). Sailing From the Seas of Chaos Into the Corridor of Stability: Practical Recommendations to Increase the Informational Value of Studies. Perspectives on Psychological Science, 9(3), 278–292. https://doi.org/10.1177/1745691614528520

Nakagawa, S., & Schielzeth, H. (2013). A general and simple method for obtaining R 2 from generalized linear mixed-effects models. Methods in Ecology and Evolution, 4(2), 133–142. https://doi.org/10.1111/j.2041-210x.2012.00261.x

7. PLOS authors have the option to publish the peer review history of their article (what does this mean?). If published, this will include your full peer review and any attached files.

Reviewer #2: No

Reviewer #3: No

---

## [Author Response · Author response to Decision Letter 1]

23 Apr 2024

PONE-D-23-06734R1

Facial mimicry interference reduces working memory accuracy for facial emotion expressions when task load is low but not when it is high

PLOS ONE

Dear Dr. Holmer,

Thank you for submitting your manuscript to PLOS ONE. After careful consideration, we feel that it has merit but does not fully meet PLOS ONE’s publication criteria as it currently stands. Therefore, we invite you to submit a revised version of the manuscript that addresses the points raised during the review process.

Thank you for your patience while we collected the reviewer's feedback. As one of the original reviewers was unable to accept the revision task, a third reviewer (R3) was brought on to provide a new review. The original Reviewer 1 was happy with your revisions and is ready to accept the manuscript. However, R3 has asked for revisions. Most of these relate to statistical reporting. While they do not require the collection of new data, they are significant enough to be classified as major revisions. Therefore, I invite you to revise and resubmit the manuscript to address R3's concerns.

Response: Thank you for the opportunity to resubmit another version of the manuscript. We have carefully considered all comments from R3 and tried to do our best to adhere to their requests. However, because of the statistical model we use (generalized linear mixed model), we do not see a straightforward approach to estimate an MDES and effect size metrics for our fixed effects on comparable scales (see discussions on standardized effect size metrics for glmer in e.g. Rights & Sterba, 2019), which means that we do not think we can do exactly what R3 is asking for. As noted by you in the previous round of reviews, power estimation in glmer models should be approached by using simulations (as implemented in e.g. the simr package, Green & MacLeod, 2016), and this should be done prior to data collection. R3 also indicated that this could be a useful tool for us to estimate effect sizes for fixed effects. 

Using the simr and mixedpower packages (following Kumle et al.’s, 2021, tutorial), it is possible to estimate power for a range of effects on the same scale (unstandardized) as the output we get from the glmer model we have used in the manuscript. Using this approach to estimate power for a range of effects given the design we have, seems to be a way to get something along the lines of what R3 is asking for. Thus, to test the precision of our design, we implemented Kumle et al.’s method to estimate power for a range of unstandardized effects of the critical interaction term (starting at -.10, and moving down to -.50 in -.05 increments), keeping the random effect structure and the other fixed effects constant (but excluding the covariates, since these were not part of the original design). The simulations included the generation of an artificial glmer model (using the makeGlmer function in the simr package), based on the experimental design our study have (2 levels of load, 3 levels of facial mimicry interference, with 54 observations in each load by interference combination, and 36 participants = 11664 observations) and using our observed fixed effects as a starting point. Then, we simulated the power of the design for the interaction term by varying the size of it in the steps defined above (from -.10, to -.50). In the graph below, we have fitted a power curve based on the output from our simulations (the betas are plotted as absolute values, so that the curve extends to the right instead of to the left). For a beta of .30 (-.30 in the simulation), we see that the power approaches 80% (76% is the estimate from the simulation). This is just a slightly higher (in absolute terms) value than the observed effect of the interaction in the results we get, which we think suggest that the precision of the model fine (note that the size of the effects from the simulations cannot be directly compared to the size of the effects of the covariates in the model reported in the manuscript, since these are on another scale compared to the factors). For now, we do not see a strong motivation for including the (post hoc) power simulations in the manuscript, but if you or R3 find this useful, we can of course reconsider.

[See attached "Response to reviewers" word-file for the figure, I was not able to include it here]

We also noted in the comments from R3 that they have the impression that we draw to strong conclusions based on the interaction we see. We thought that we had soften our claims in the previous round of reviews, but it became clear to us when re-reading the manuscript once again that we had not. One obvious example of this, is that we in the reporting of the results did not comment on the unexpected direction of the interaction effect. Instead, we highlighted the interaction as our most important finding. Our prediction, as stated in the introduction, was in the other direction (lines 151-153: “We predicted that performance accuracy would decrease with increasing working memory load. We expected this effect to be the strongest when facial mimicry interference was high”). That the direction of the interaction was unexpected is now clarified in the results, the discussion, and the conclusions, and we have de-emphasized the impact of this finding:

o We have changed the title again, now it reads: 

“Facial mimicry interference reduces working memory accuracy for facial emotion expressions when task load is low” (we have deleted “...but not when it is high”)

o The emphasis on the lack of an effect of mimicry interference at high load in the conclusion has been toned down. In the abstract: 

“We conclude that facial mimicry might support working memory for emotion expressions when task load is low, but that the supporting effect possibly is reduced when the task becomes more cognitively challenging.” (p. 2)

In the first sentence of the Conclusion:

Was before “Facial mimicry might influence the precision of representations of facial emotion expressions when load on working memory is low but perhaps not when load is high”, reads now “Facial mimicry might influence the precision of representations of facial emotion expressions when load on working memory is low”. (p. 24)

o We emphasize more clearly that the direction of the interaction was unexpected. 

In the Abstract:

Was before “…the high level of mimicry interference impaired accuracy when working memory load was low (1-back) but not when load was high (2-back)”, now reads “…the high level of mimicry interference impaired accuracy when working memory load was low (1-back) but, unexpectedly, not when load was high (2-back)”. (p. 2)

In the Results:

Was before “However, the most important finding was that the interaction between working memory load and high interference was statistically significant…”, now reads “However, the critical test of our study was the interaction between working memory load and high interference. Surprisingly, the statistically significant interaction was in the opposite direction of our prediction…”. (p. 15)

And in the Discussion:

Was before “We further cautiously propose that a general principle of the neurocognitive system is that when working memory demands increase, the system responds by filtering out potentially distracting information”, now reads “To explain the unexpected finding that working memory load seems to suppress the negative effect of facial mimicry on precision, we cautiously propose that a general principle of the neurocognitive system is that when working memory demands increase, the system responds by filtering out potentially distracting information” (p. 20)

Was before “This could be due to the relatively small sample or features of the task design”, now reads “It should be acknowledged that this surprising result could be due to some characteristics of the present sample or features of the task design, possibly limiting the precision of the statistical model even though the total number of observations per condition was high (36 participants, and 54 trials per load by mimicry interference condition)”. (p. 22) 

We hope that these changes make it clear that we do not intend to oversell anything here, we are just trying to make sense out of an unexpected, and potentially very interesting, finding.

In addition to the changes described above, we have we have followed R3’s suggested changes in points 2-4.

References

Green, P., & MacLeod, C. J. (2016). SIMR: An R package for power analysis of generalized linear mixed models by simulation. Methods in Ecology and Evolution, 7(4), 493–498. https://doi.org/10.1111/2041-210X.12504

Kumle, L., Võ, M. L. H., & Draschkow, D. (2021). Estimating power in (generalized) linear mixed models: An open introduction and tutorial in R. Behavior Research Methods, 53(6), 2528–2543.

Rights, J. D., & Sterba, S. K. (2019). Quantifying explained variance in multilevel models: An integrative framework for defining R-squared measures. Psychological Methods, 24(3), 309–338. https://doi.org/10.1037/met0000184

We look forward to receiving your revised manuscript.

Kind regards,

Steven R Livingstone

Academic Editor

PLOS ONE

Reviewers' comments:

Reviewer's Responses to Questions

Comments to the Author

1. If the authors have adequately addressed your comments raised in a previous round of review and you feel that this manuscript is now acceptable for publication, you may indicate that here to bypass the “Comments to the Author” section, enter your conflict of interest statement in the “Confidential to Editor” section, and submit your "Accept" recommendation.

Reviewer #2: All comments have been addressed

Reviewer #3: (No Response)

2. Is the manuscript technically sound, and do the data support the conclusions?

Reviewer #2: Yes

Reviewer #3: Partly

3. Has the statistical analysis been performed appropriately and rigorously? 

Reviewer #2: Yes

Reviewer #3: Yes

4. Have the authors made all data underlying the findings in their manuscript fully available?

Reviewer #2: Yes

Reviewer #3: Yes

5. Is the manuscript presented in an intelligible fashion and written in standard English?

Reviewer #2: Yes

Reviewer #3: Yes

6. Review Comments to the Author

Reviewer #2: (No Response)

Reviewer #3: In the revised manuscript titled “Facial mimicry interference reduces working memory accuracy for facial emotion expressions when task load is low but not when it is high,” the authors investigated the effect of blocking spontaneous facial mimicry on working memory recall for facial emotion expressions. They reported that high-level mimicry interference was effective for 1-back but not for 2-back recalls. The authors made great efforts in addressing comments from the editor and reviewers 1 & 2. However, key points about the study’s affordance to provide enough precision for their observed effects were not addressed.

1. The authors stated that they have followed Lakens’ suggestions (Lakens, 2022) in sample size justification. While Lakens provided six principles, the authors only stated reasons for resource limitations and heuristics. I want to emphasize that the authors need to demonstrate whether their present study provided precision in effect estimation for what they expected to observe and what they actually observed. The issue of precision in effect estimation is demonstrated in (Lakens & Evers, 2014) Table 1. Small effects require a large sample size to reach the point of stability in effect estimation. With an insufficient sample size, the observed effects are likely unreliable and unlikely to be reproduced. Therefore, I would like to request explicitly:

a. A sensitivity analysis to determine, at the current design and sample size, what the minimally detectable effect size (MDES) is with 80% power. Note that this has nothing to do with your observed effect size. In Lakens’ terms (Lakens, 2022), this is “which minimal effect size will be statistically significant.”

b. Reports of observed effect sizes for all statistical results (Green & MacLeod, 2016; Nakagawa & Schielzeth, 2013).

c. Considering whether the present study afforded enough precision in effect estimation for the significant and null results they reported in the manuscript. Suppose the effect size reported is much smaller than the present study's precision. In that case, the results reported should not be considered conclusive, and I will have difficulty recommending publishing this study as it is.

Response

- In the previous round of revisions, we tried to make it clear that we do not see our results as conclusive – in fact, we were rather surprised by the direction of the interaction. At the start, we expected that the effect of mimicry would be stronger

---

## [Decision Letter · Decision Letter 2]

20 May 2024

PONE-D-23-06734R2Facial mimicry interference reduces working memory accuracy for facial emotion expressionsPLOS ONE

Dear Dr. Holmer,

Thank you for submitting your manuscript to PLOS ONE. I have now received reviews from two expert reviewers. As you will see, the reviews are positive, with reviewer 1 opting to accept and reviewer 2 raising only minor concerns.  For context, I was lucky secure Reviewer 2 for your revision. However, as Reviewer 1 needed to decline due to personal issues, a new Reviewer 1 was located.

We look forward to receiving your revised manuscript.

Kind regards,

Steven R Livingstone

Academic Editor

PLOS ONE

Journal Requirements:

Reviewers' comments:

Reviewer's Responses to Questions

**Comments to the Author**

1. If the authors have adequately addressed your comments raised in a previous round of review and you feel that this manuscript is now acceptable for publication, you may indicate that here to bypass the “Comments to the Author” section, enter your conflict of interest statement in the “Confidential to Editor” section, and submit your "Accept" recommendation.

Reviewer #2: All comments have been addressed

Reviewer #3: (No Response)

2. Is the manuscript technically sound, and do the data support the conclusions?

Reviewer #2: Yes

Reviewer #3: Yes

3. Has the statistical analysis been performed appropriately and rigorously? 

Reviewer #2: Yes

Reviewer #3: Yes

4. Have the authors made all data underlying the findings in their manuscript fully available?

Reviewer #2: Yes

Reviewer #3: Yes

5. Is the manuscript presented in an intelligible fashion and written in standard English?

Reviewer #2: Yes

Reviewer #3: Yes

6. Review Comments to the Author

Reviewer #2: Overall, the authors have addressed the bulk of comments raised by R3 in the last review and the tempering of conclusions and clarifications improve the manuscript.

One item to note is that R3 requested that the authors further present a sensitivity analysis based on their sample size and discuss their precision in effect estimation. The authors argue that they are unaware of a mechanism to accurately address this request for their analytic model and provide a lengthy consideration of this item in their response letter. To speak to the sensitivity, they present a power curve based on simulations for a similar model (without covariates) in the response, but this not in the manuscript. While this may be harnessed to offer a rough estimate of an MDES, the values are not comparable given the unstandardized betas reported. An estimate and reporting of standardized effects might address this issue. Alternatively, I wonder if providing an estimate based on a more basic model (i.e., ANCOVA vs. a mixed model) could at least provide a conservative estimate (given the MLM here is more powered). Regardless, the authors have addressed practical reasons for their sample size and tempered the strength of their interpretations accordingly.

As a minor point, in the new sentence on line 410-411, I believe the authors may have meant to say “…to suppress the negative effect of facial mimicry INTERFERENCE on precision.”

Reviewer #3: Again, the authors made respectable efforts to improve the coherence and transparency of their work. I have only three suggestions.

Major points:

1. Please check whether the r2glmm package (https://cran.r-project.org/web/packages/r2glmm/readme/README.html) (Jaeger et al., 2017) is applicable in providing standardized effect size estimation (semi-partial R2) for fixed effects in GLMM. If applicable, please add the standardized effect size information for all reported effects accordingly. If not, please provide the rationale, which could be judged by the editor, and I will have no further requests on this issue. The authors' simulation using simr was independent of the observed effect size, and I consider this approach valid.

Minor points:

2. Ensuring the R code version on the OSF depository is up to date is crucial for the replicability of your analysis in its latest form. I kindly request you to prioritize this update.

3. Please consider making the review process public during acceptance (I appreciate that PLoS One provides this option). This will provide the reader with the context for why the authors conducted additional analysis or simulation.

Jaeger, B. C., Edwards, L. J., Das, K., & Sen, P. K. (2017). An R 2 statistic for fixed effects in the generalized linear mixed model. Journal of Applied Statistics, 44(6), 1086–1105. https://doi.org/10.1080/02664763.2016.1193725

7. PLOS authors have the option to publish the peer review history of their article (what does this mean?). If published, this will include your full peer review and any attached files.

Reviewer #2: No

Reviewer #3: No

---

## [Author Response · Author response to Decision Letter 2]

3 Jun 2024

Dear Dr. Holmer,

Thank you for submitting your manuscript to PLOS ONE. I have now received reviews from two expert reviewers. As you will see, the reviews are positive, with reviewer 1 opting to accept and reviewer 2 raising only minor concerns. For context, I was lucky secure Reviewer 2 for your revision. However, as Reviewer 1 needed to decline due to personal issues, a new Reviewer 1 was located.

- Thanks for the opportunity for another resubmission. We are glad to see that the reviewers overall seem happy with the manuscript. To deal with the last few comments, we have done the following:

1) There was a general comment related to “Journal requirements” about the references, and for us to check whether any study we are referring to has been retracted. We checked doi-numbers for the references against Retraction Watch and based on that search, which resulted in 0 references being identified as a retracted study, we conclude that we can keep all references. The reference list was updated with two minor changes, i.e., corrected doi-numbers for two references.

2) In the last round of reviews, one proposed change was for us to add a sensitivity analysis by comparing MDES to some standardized effect size metric for the effects in our glmer model. We concluded that this probably is not possible with available statistical tools. As a response to this, Rev 3 proposed for us to try the r2glmm package to estimate R2-like effect sizes for individual effects. From what we understand this does not offer a viable solution. Firstly, the documentation of this package states that the r2beta function that is used to estimate effect sizes for individual predictors does not work with glmer models from lme4 (see https://cran.r-project.org/web/packages/r2glmm/r2glmm.pdf). Secondly, when we run this function on our model we get an error (In calc_sgv(nblocks = nclusts, blksizes = obsperclust, vmat = SigHat) : Some SGV estimates are non-finite and have been adjusted) and some spurious results:

> r2beta(model = model, method = 'sgv', data = data) 

Effect Rsq upper.CL lower.CL

1 Model 0.049 0.057 0.042

2 Load 0.017 0.022 0.012

5 valence 0.004 0.006 0.002

6 order 0.002 0.004 0.001

7 arousal 0.001 0.003 0.000

3 High 0.001 0.002 0.000

8 Load:High 0.000 0.002 0.000

4 Low 0.000 0.000 0.000

9 Load:Low 0.000 0.000 0.000

Thirdly, even if we assume that the output is valid, we do not see how we could estimate a standardized MDES on the same scale as these R2-like estimates, since R2 estimates for linear models are not comparable. Thus, we will not be able to discuss the sensitivity of our model based on this.

In the last round of reviews, we provided a simulation-based power analysis as an alternative approach to assess the sensitivity of the model. The downside with this approach is that it is based on unstandardized estimates that might not be easy to understand outside the context of the study. However, the unstandardized effects can be directly compared to the output we get from our glmer model. We lean towards that this might be the better approach for our study. In the revised version of the manuscript, we have added a short section discussing the results from our simulations in relation to the effects we see in the study, and we also attach a description of the simulations as an Appendix. The new text included at p. 16-17, lines 356-369, reads as follows:

“Since no proper power analysis was performed prior to the study, a sensitivity analysis based on post hoc power simulations was conducted. Simulations followed the guidelines from Kumle et al. [75], and we only considered the main manipulations of the experiment (working memory load and facial mimicry interference), which means that the simulation results can only be compared to the effects of those factors in the results from the generalized linear mixed effects model reported above. All effects but the interaction between high interference and working memory load (which was the critical test in the design) were set to a fixed value. The overall pattern of the simulation results suggested that beta weights larger than approximately .30 could be detected with at least 80% power, whereas beta weights in a lower range (.10-.20) revealed a power of less than 50% (for more details, see S1 Appendix). Thus, we cannot fully reject the possibility that poor sensitivity of our design might explain why we did not observe an effect of high interference when working memory load was high.”

We look forward to receiving your revised manuscript.

Kind regards,

Steven R Livingstone

Academic Editor

PLOS ONE

Journal Requirements:

- We have checked all references’ doi-numbers against Retraction Watch, and the results indicate that none of the studies we reference have been retracted. In the previous version, the doi-number for two references was incorrect/missing and has now been corrected/added (Rönnberg et al., 2022 & Dimberg et al., 2000).

Reviewers' comments:

Reviewer's Responses to Questions

Comments to the Author

1. If the authors have adequately addressed your comments raised in a previous round of review and you feel that this manuscript is now acceptable for publication, you may indicate that here to bypass the “Comments to the Author” section, enter your conflict of interest statement in the “Confidential to Editor” section, and submit your "Accept" recommendation.

Reviewer #2: All comments have been addressed

Reviewer #3: (No Response)

2. Is the manuscript technically sound, and do the data support the conclusions?

Reviewer #2: Yes

Reviewer #3: Yes

3. Has the statistical analysis been performed appropriately and rigorously? 

Reviewer #2: Yes

Reviewer #3: Yes

4. Have the authors made all data underlying the findings in their manuscript fully available?

Reviewer #2: Yes

Reviewer #3: Yes

5. Is the manuscript presented in an intelligible fashion and written in standard English?

Reviewer #2: Yes

Reviewer #3: Yes

6. Review Comments to the Author

Reviewer #2: Overall, the authors have addressed the bulk of comments raised by R3 in the last review and the tempering of conclusions and clarifications improve the manuscript.

One item to note is that R3 requested that the authors further present a sensitivity analysis based on their sample size and discuss their precision in effect estimation. The authors argue that they are unaware of a mechanism to accurately address this request for their analytic model and provide a lengthy consideration of this item in their response letter. To speak to the sensitivity, they present a power curve based on simulations for a similar model (without covariates) in the response, but this not in the manuscript. While this may be harnessed to offer a rough estimate of an MDES, the values are not comparable given the unstandardized betas reported. An estimate and reporting of standardized effects might address this issue. Alternatively, I wonder if providing an estimate based on a more basic model (i.e., ANCOVA vs. a mixed model) could at least provide a conservative estimate (given the MLM here is more powered). Regardless, the authors have addressed practical reasons for their sample size and tempered the strength of their interpretations accordingly.

- Thanks for acknowledging our efforts to deal with the comments in the last round of reviews. Regarding the sensitivity analysis, see our response to Rev 3 below.

As a minor point, in the new sentence on line 410-411, I believe the authors may have meant to say “…to suppress the negative effect of facial mimicry INTERFERENCE on precision.”

- Thanks for noticing this. Corrected in the revised version.

Reviewer #3: Again, the authors made respectable efforts to improve the coherence and transparency of their work. I have only three suggestions.

Major points:

1. Please check whether the r2glmm package (https://cran.r-project.org/web/packages/r2glmm/readme/README.html) (Jaeger et al., 2017) is applicable in providing standardized effect size estimation (semi-partial R2) for fixed effects in GLMM. If applicable, please add the standardized effect size information for all reported effects accordingly. If not, please provide the rationale, which could be judged by the editor, and I will have no further requests on this issue. The authors' simulation using simr was independent of the observed effect size, and I consider this approach valid.

- Thanks for your effort in trying to find a viable approach to standardize effect sizes for the effects in our model. From what we understand, using the r2beta function from r2glmm to get R2-like estimates for individual predictors in glmer models is an experimental approach, and for models built with the lm4 package the package documentation says the function currently only can be used with linear mixed models (see, https://cran.r-project.org/web/packages/r2glmm/r2glmm.pdf). Indeed, when running the function on the model, we see some spurious results (tiny estimates for some of the statistically significant effects, see below) and a warning message (In calc_sgv(nblocks = nclusts, blksizes = obsperclust, vmat = SigHat) : Some SGV estimates are non-finite and have been adjusted). If we assume that this function provides valid estimates, the alternative interpretation would be that some of the effects we see are indistinguishable from zero. However, this package (and the earlier packages we have tried) does not seem to be a viable solution, especially not if the goal is to compare to MDES based on a more traditional statistical approach (since pseudo-R2 metrics are not comparable to such estimates). 

Output from R:

 > r2beta(model = model, method = 'sgv', data = d) 

Effect Rsq upper.CL lower.CL

1 Model 0.049 0.057 0.042

2 Load 0.017 0.022 0.012

5 valence 0.004 0.006 0.002

6 order 0.002 0.004 0.001

7 arousal 0.001 0.003 0.000

3 High 0.001 0.002 0.000

8 Load:High 0.000 0.002 0.000

4 Low 0.000 0.000 0.000

9 Load:Low 0.000 0.000 0.000

Taken everything discussed in the previous and the current round of reviews into account, leads us to draw the conclusion that the simulation-based approach is preferrable over the other solutions. Thus, we have added a paragraph to discuss the results from the simulations to the paper (p. 16-17, lines 356-369):

“Since no proper power analysis was performed prior to the study, a sensitivity analysis based on post hoc power simulations was conducted. Simulations followed the guidelines from Kumle et al. [75], and we only considered the main manipulations of the experiment (working memory load and facial mimicry interference), which means that the simulation results can only be compared to the effects of those factors in the results from the generalized linear mixed effects model reported above. All effects but the interaction between high interference and working memory load (which was the critical test in the design) were set to a fixed value. The overall pattern of the simulation results suggested that beta weights larger than approximately .30 could be detected with at least 80% power, whereas beta weights in a lower range (.10-.20) revealed a power of less than 50% (for more details, see S1 Appendix). Thus, we cannot fully reject the possibility that poor sensitivity of our design might explain why we did not observe an effect of high interference when working memory load was high.”

The details of the simulation are added to a new supplement (S1 Appendix), and the scripts on OSF have been updated to include the simulations as well. 

Minor points:

2. Ensuring the R code version on the OSF depository is up to date is crucial for the replicability of your analysis in its latest form. I kindly request you to prioritize this update.

- We apologize that we missed to update the code on OSF when we changed the analyses. We have now ensured that the data and code available on OSF is up to date, including the code for the power simulations.

3. Please consider making the review process public during acceptance (I appreciate that PLoS One provides this option). This will provide the reader with the context for why

---

## [Decision Letter · Decision Letter 3]

12 Jun 2024

Facial mimicry interference reduces working memory accuracy for facial emotion expressions

PONE-D-23-06734R3

Dear Dr. Holmer,

We’re pleased to inform you that your manuscript has been judged scientifically suitable for publication and will be formally accepted for publication once it meets all outstanding technical requirements.

Kind regards,

Steven R Livingstone

Academic Editor

PLOS ONE

Additional Editor Comments (optional):

Reviewers' comments:

Reviewer's Responses to Questions

**Comments to the Author**

1. If the authors have adequately addressed your comments raised in a previous round of review and you feel that this manuscript is now acceptable for publication, you may indicate that here to bypass the “Comments to the Author” section, enter your conflict of interest statement in the “Confidential to Editor” section, and submit your "Accept" recommendation.

Reviewer #3: All comments have been addressed

2. Is the manuscript technically sound, and do the data support the conclusions?

Reviewer #3: Yes

3. Has the statistical analysis been performed appropriately and rigorously? 

Reviewer #3: Yes

4. Have the authors made all data underlying the findings in their manuscript fully available?

Reviewer #3: Yes

5. Is the manuscript presented in an intelligible fashion and written in standard English?

Reviewer #3: Yes

6. Review Comments to the Author

Reviewer #3: (No Response)

7. PLOS authors have the option to publish the peer review history of their article (what does this mean?). If published, this will include your full peer review and any attached files.

Reviewer #3: No

---

## [Editor Report · Acceptance letter]

17 Jun 2024

PONE-D-23-06734R3 

PLOS ONE

Dear Dr. Holmer, 

I'm pleased to inform you that your manuscript has been deemed suitable for publication in PLOS ONE. Congratulations! Your manuscript is now being handed over to our production team.

Kind regards, 

on behalf of

Dr. Steven R Livingstone 

Academic Editor

PLOS ONE